# The Hidden Pandemic of COVID-19-Induced Organizing Pneumonia

**DOI:** 10.3390/ph15121574

**Published:** 2022-12-16

**Authors:** Evgeny Bazdyrev, Maria Panova, Valeria Zherebtsova, Alexandra Burdenkova, Ivan Grishagin, Fedor Novikov, Vladimir Nebolsin

**Affiliations:** 1Research Institute for Complex Issues of Cardiovascular Diseases, 6, Sosnoviy Blvd., 650002 Kemerovo, Russia; 2Pharmenterprises LLC, Skolkovo Innovation Center, Bolshoi Blvd., 42(1), 143026 Moscow, Russia; 3Zelinsky Institute of Organic Chemistry, Russian Academy of Sciences, 47 Leninsky Avenue, 119991 Moscow, Russia; 4Dmitry Mendeleev University of Chemical Technology of Russia, Miusskaya sq. 9, 125047 Moscow, Russia; 5Rancho BioSciences, 16955 Via Del Campo Suite 200, San Diego, CA 92127, USA

**Keywords:** COVID-19 sequelae, organizing pneumonia, SARS-CoV-2 pneumonia, corticosteroids

## Abstract

Since the beginning of the COVID-19 pandemic, clinical, radiological, and histopathological studies have provided evidence that organizing pneumonia is a possible consequence of the SARS-CoV2 infection. This post-COVID-19 organizing pneumonia (PCOP) causes persisting dyspnea, impaired pulmonary function, and produces radiological abnormalities for at least 5 weeks after onset of symptoms. While most patients with PCOP recover within a year after acute COVID-19, 5–25% of cases need specialized treatment. However, despite substantial resources allocated worldwide to finding a solution to this problem, there are no approved treatments for PCOP. Oral corticosteroids produce a therapeutic response in a majority of such PCOP patients, but their application is limited by the anticipated high-relapse frequency and the risk of severe adverse effects. Herein, we conduct a systematic comparison of the epidemiology, pathogenesis, and clinical presentation of the organizing pneumonias caused by COVID-19 as well as other viral infections. We also use the clinical efficacy of corticosteroids in other postinfection OPs (PIOPs) to predict the therapeutic response in the treatment of PCOP. Finally, we discuss the potential application of a candidate anti-inflammatory and antifibrotic therapy for the treatment of PCOP based on the analysis of the latest clinical trials data.

## 1. Introduction

Over the past two years, the world has been tracking the progress of the COVID-19 pandemic using the number and severity of cases [1] and the number of deaths [2]. Multiple COVID-19 variants have been discovered. Among the variants of concern, the Alpha variant was identified first, followed by the Beta, Gamma, Delta, and Omicron variants. In complete agreement with the known patterns of the viral evolution, each subsequent variant was less virulent and more transmissible than its predecessor. Furthermore, it has been reported that mass vaccination resulted in vaccine-escape mutations becoming the dominant viral evolutionary mechanism in highly vaccinated populations [3]. The Omicron variant, which became dominant in 2021, thus carries the lowest risk of hospitalization but also demonstrated the greatest propensity to mutate. Early in the pandemic, it was discovered that the SARS-CoV-2 genome is almost 80% identical to the sequence of SARS-CoV-1 that caused the 2003 epidemic. From the genome similarity standpoint, this is a large evolutionary gap, which also means that if any of the SARS-CoV-1 therapeutic targets turn out to be relevant for SARS-CoV-2, they would be sufficiently robust and generic to be relevant for all variants thereof. Recent advances in management of COVID-19 allow to focus on the long-term physical sequelae of acute infection, which are becoming a “hidden pandemic” in the Western world [4,5]. As of October 2022, the incidence of persistent post-COVID-19 symptoms exceed 17 million cases in Europe alone based on the WHO estimate [6]. Despite the strong unmet need, only symptomatic treatments for these COVID-19 sequelae are available.

The clinical presentation of COVID-19 sequelae, in general, is characterized by the presence of interstitial lung damage associated with impaired respiratory function and accompanied by respiratory symptoms, including shortness of breath, chest pain, and dry cough [5,6]. The post-COVID-19 radiological imaging patterns typically comprise a range of ground-glass opacities from focal unilateral to diffuse bilateral that have progressed to or coexisted with consolidations. Importantly, such imaging patterns have been found even after asymptomatic COVID-19 [7] and are typical for organizing pneumonia (OP) [8].

Organizing pneumonia (OP) is one of the most common lung lesions in many infectious diseases [9,10,11,12]. Virus-induced damage to alveolar epithelial cells (pneumocytes) and endothelial cells associated with an active inflammatory process seems to be the key etiological factor in OP, as well as post-COVID-19 pulmonary sequelae (PCPS). Abnormal chest auscultation, reduced diffusing capacity of the lung, and slow progressive respiratory failure are common both for OP and PCPS [13,14]. The clinical and radiological findings are frequently nonspecific, and a firm diagnosis cannot be established without a lung biopsy [15]. Similarities in etiology, pathogenesis, and clinical manifestations of these two conditions justify the use of common approaches to symptomatic and pathogenetic therapy [16,17,18].

Treatment with oral corticosteroids results in complete or partial recovery in up to 80% of OP cases within a few weeks to 3 months [19,20]. For PCPS, a similar response has been demonstrated in small exploratory trials [16,17,18]. Unfortunately, after one-year follow-up, OP relapses were reported for 40% of responders along with a 9.4% mortality [13]. Herein, we aim to demonstrate that there is an unprecedented demand for novel OP therapies that are safe and effective due to the high risk of reinfection and the prevalence of OP-like PCPS. The effective management of organizing pneumonia may become the key challenge in drug-discovery in the next few years.

## 2. Time-Dependent Epidemiology of Post-COVID-19 Organizing Pneumonia (PCOP)

According to several clinical trials, the severity and the duration of the COVID-19 infection are the key factors determining the prevalence of OP-like PCPS [21,22] (Appendix A).

At 3 months after COVID-19-related hospitalization, about 40–50% of patients suffer from impaired pulmonary gas exchange, and the prevalence of lung lesions reaches 65% [23,24,25]. The development of the OP-like PCPS is less typical in patients who had a moderate form of COVID-19: at 3 months after the diagnosis, gas exchange disorders are observed in only 10–20% of patients (Figure 1) [23,24]. Of patients with impaired pulmonary gas exchange, 50–70% have respiratory disease symptoms, and 70% have damage to the lung parenchyma [22,26]. Thus, *at least half* of patients with impaired pulmonary gas exchange have residual lung lesions and clinical manifestations of OP.

Accelerated COVID-19 vaccine development and mass vaccination in first half of 2021, have been reported to substantially reduce mortality and the rate of critical cases. Besides that, some studies suggest that at least some COVID-19 vaccines may prevent, mitigate, or even reverse existing long-COVID-19 symptoms. For example, it was reported that vaccines reduce the risk and burden of long-COVID-19 symptoms in 28 percent of the over-60 age group [27]. In the younger age groups, COVID-19 vaccines reduce the risk of long-COVID-19 symptoms twofold in outpatients [28] and three times in those who develop severe COVID-19 [29], but the strength of the evidence is low [28].

Similar effect on long-COVID-19 prevalence has been observed for the Omicron compared with the Delta cases. After Omicron infection, the likelihood of developing symptoms is four times lower in unvaccinated patients and two times lower in the case of those who got vaccinated within 180 days before the disease [30,31]. Thus, as of October 2022, the probability of persisting OP-like PCPS within 3 months after hospitalization is about 5% and just over 1% for moderate COVID-19.

Given that the proportion of severely ill patients (dyspnea, hypoxia, or >50% pulmonary involvement) is 17%, and the proportion of critically ill patients is 2% (respiratory failure, shock, or multiple-organ-dysfunction syndrome) [32], it is possible to estimate the annual number of new OP-like PCPS cases in the United States (990,000), the UK (280,000), and Germany (545,000) (Figure 2). The extrapolation of these data from developed countries provides the approximate annual global incidence of 7 million of new OP-like PCPS cases.

The number of patients hospitalized with confirmed COVID-19 infection and patients experiencing OP-like symptoms after three months of hospitalization was calculated using ourworldindata.org dataset (accessed on 17 October 2022). The total percentage of survived patients was calculated taking into account the different severity of the disease. The probability of developing OP-like symptoms was 40%, 25%, and 15% for critical, severe, and moderate patients, respectively (see above). Hospitalized patients were considered severe, and those who needed ICU during hospitalization were considered critical. Moderate patients were calculated from the number of total cases minus the hospitalized, adjusted for mild patients, which were 81% according to [33]. Mortality rates were 33%, 17%, and 5% for critical, severe, and moderate COVID-19 patients, respectively [34,35]. Finally, a reduction in the probability of developing long-COVID-19 for vaccinated patients was taken into account (50% for nonhospitalized and 33% for hospitalized) and for patients recovering from the Omicron variant (25% for nonvaccinated and 25% for vaccinated) (see above).

It appears that the spontaneous recovery of the respiratory function takes place during the first 6 to 9 months after the acute phase. During this period, a complete recovery from moderate COVID-19 may occur [36], but OP-like disorders persist in 20% of severe and 40% of critical cases [21,37,38,39]. Furthermore, in most cases, the restoration of respiratory function stops almost completely at the 6-month mark, so the prevalence of persistent gas exchange disorders among hospitalized patients is approximately 25% [38,40,41,42] (see Figure 1). Over the past 2 years alone (January 2020 to December 2021), approximately 10 million patients in the US have been hospitalized for COVID-19, suggesting that 300,000 patients will develop sustained, if not permanent, sequelae. Worldwide, since the start of the pandemic, the number of people with persistent lesions of the lung parenchyma associated with persistent impairment of the pulmonary gas exchange and concomitant respiratory symptoms can be estimated at 800,000.

## 3. Pathogenesis and Clinical Manifestation of Post-Infection OP (PIOP)

Persistent inflammation of the lung tissue appears to be the key etiological factor in PIOP and OP-like PCPS (Table 1). Inflammation is a consequence of aberrant immune response to the virus-induced damage to alveolar epithelial cells (pneumocytes) and endothelial cells [43]. The prominent presence of an inflammatory process is confirmed by laboratory testing, which demonstrates an increased content of neutrophils [44], proinflammatory mediators (IL-6, IL-8, TNF-α), and active inflammatory markers (CRP, ferritin) in the blood of patients who have had viral infection and are suffering from OP-like sequelae [44,45,46,47,48]. Proinflammatory mediators and chemoattractants stimulate the homing of macrophages and neutrophils to the lungs [45]. Accumulation of neutrophils in blood capillaries and interalveolar septa leads to intra-alveolar edema [49], avalanche ROS production, and further damage to the lung parenchyma, alveolar epithelial, and vascular endothelial cells (Figure 3, Table 1). A similar mechanism of lung damage occurs in other OPs [50] and specifically PIOP [51].

Capillary endothelial cells and alveolar epithelial cells injury leads to the leakage of plasma protein, and, in particular, coagulation factors [52]. The subsequent intra-alveolar protein coagulation produces fibrin deposits on alveolar surfaces [52]. Thus, vascular damage, such as capillary congestion, intracapillary microthrombosis, alveolar hemorrhage, and interstitial and intra-alveolar edema frequently has contemporaneous association with desquamation and reactive hyperplasia of type II pneumocytes, hyaline membrane formation, and interstitial inflammatory response [52].

Active inflammatory processes in the respiratory tract tissues resulting from tissue damage and alveolar hemorrhage significantly impair pulmonary gas exchange [52]. Moreover, persistent anemia after acute infection and residual lung damage associated with a right-to-left intrapulmonary shunt can also play a role in post-COVID-19 pulmonary gas exchange abnormalities [92,93]. Altered gas-exchange function leads to hypoxemia and raises pulmonary vascular resistance by widespread hypoxic pulmonary vasoconstriction [94]. Pulmonary hypertension promotes vascular leakage, accompanied by alveolar hemorrhage, and interstitial and intra-alveolar edema [52].

The gaps in the injured epithelial basal membranes facilitate the migration of the fibroblasts from the alveolar interstitium and the subsequent colonization of the alveolar space [95]. The same process takes place in OP [96]. Proliferating fibroblasts then differentiate into myofibroblasts [97], producing the extracellular myxoid matrix [98]. Inflammation ensues as inflammatory cells flood the alveolar interstitium [53].

Finally, the tissue repair process begins. First, type II pneumocytes multiply to fill the gaps in the epithelial lining of the basal membranes. After that, the intra-alveolar granulation tissue undergoes reorganization into mature fibrotic collagen-rich bundles or “buds.” Those gradually fill the alveoli, alveolar ducts, and distal bronchioles. The overall parenchymal architecture remains unaffected [53].

According to laboratory tests, there is an increased content of tissue damage markers (e.g., lactate dehydrogenase) in the blood of the patients who have had an infection and suffer from OP-like sequelae [45]. Radiological presentation consists focal unilateral to diffuse bilateral ground-glass opacities with or without consolidation or a “crazy-paving” pattern located primarily in the lower fields [99]. Ground-glass opacities are the dominant finding regardless of the disease stage; however, consolidations and crazy-paving patterns are generally associated with severe illness [100,101]. The formation of the reticular pattern might be associated with the interstitial lymphocyte infiltration causing interlobular septal thickening [102].

Only 3% of patients, typically with advanced disease [103], experience lobar consolidation, miliary pattern, lung nodules or masses, cavitation, or pleural effusion [104]. Spontaneous resolution of the secondary OP within a year after the infection is observed in 60–80% of patients. In the remaining 20–40% of patients, gradual interlobular and intralobular septal thickening is observed. These findings are associated with bronchial dilations and are indicative of interstitial disease, which, in turn, suggests the development of fibrosis [99,105].

Persistent respiratory problems related to lung fibrosis may ultimately produce substantial population morbidity, long-term disability, and mortality. Worse, there is evidence that, even in mild cases, fibrosis may become one of the major long-term post-COVID-19 complications [106]. The similarities between COVID-19 and SARS pathogeneses make it possible to infer the incidence of post-COVID-19 pulmonary fibrosis using long-term observational studies of SARS-induced lung pathology [107,108]. In about 80% of SARS patients with relevant sequelae, fibrotic lung damage resolved within a year, while the other 20% of patients experienced substantial progression of the disease up to 10 years after the diagnosis [107,109,110,111,112]. Consequently, one can estimate the incidence rate of lung fibrosis developing after severe COVID-19 at 2–6%. This means that the prevalence of the post-COVID-19 lung fibrosis is likely 10–100 times higher than the risk of idiopathic lung fibrosis [113]. Currently, despite substantial resources allocated worldwide, only symptomatic treatments for post-COVID-19 pulmonary fibrosis are available. For example, a 6-month treatment with anti-inflammatory drugs following COVID-19-induced pneumonia was demonstrated to improve the impaired diffusing capacity of the lungs [113].

## 4. Pharmacological Management and Outcomes in PIOP

Despite the strong unmet need, there is no approved treatment for OP-like PCPS. Analysis of the pathogenetic features of the disease indicates the key role of the aberrant inflammatory response in the development of pathology. The most straightforward approach to therapy for OP and OP-like PCPS is the nonselective suppression of inflammation with oral corticosteroid with or without concomitant immunosuppressive treatment.

### 4.1. OP Management with Oral Corticosteroid

Steroids are generally regarded as the standard treatment for OP, and usually lead to rapid and complete recovery in most cases [70]. Steroid treatment reduces concentrations of inflammatory mediators, such as IL-1β, IL-8, and TNF-α, produced by the endothelium and respiratory tract cells. Oral corticosteroids elicit a therapeutic response in OP (assessed by clinical and radiological findings) in no more than 80–85% of patients [19,114,115,116].

The similarity of the etiology, pathogenesis, and clinical presentation of the OP-like PCPS and PIOP has become the basis for the use of oral corticosteroids for the treatment of the COVID-19 sequelae. Multicenter randomized prospective clinical trials of corticosteroid therapy for OP have not been conducted [70], so the decision on the choice of the regimen and duration of therapy can only be based on general clinical considerations, such as the rate of disease progression and its severity [115,117]. To date, numerous case reports are available describing the successful use of corticosteroids for the restoration of the parenchyma and the pulmonary function, as well as the relief of concomitant symptoms in patients who have had coronavirus infection [61,63,64,118,119,120,121,122,123]. Further, three small exploratory studies evaluating the efficacy of oral steroids in controlling the OP-like effects of COVID-19 had been completed and their data are available [16,17,18].

A single-center randomized parallel clinical trial compared high (910 mg for the whole study) and low (420 mg for the whole study) doses of prednisone in 130 patients who recovered from COVID-19 3–8 weeks prior to the study and had residual lesions affecting 20% or more of the parenchyma or residual lung lesions of any kind accompanied by severe exercise intolerance and respiratory symptoms [18]. At the end of 6 weeks of therapy, both high and low doses of prednisolone demonstrated similar effectiveness. Radiological response was observed in 80% of patients, with 23% of those expected to attain a complete response in the future. An improvement in dyspnea was observed in 90% of patients [18]. The results obtained are in good agreement with the data of two other clinical studies [16,17]. In one of those, a retrospective cohort study, the outcome of OP-like PCPS was assessed in 49 patients and prolonged oral corticosteroids have been used in 24 of those cases [17]. An initial dose of prednisolone, 36 mg once a day (QD) for 1 week, was gradually decreased by 3–12 mg weekly or biweekly. Within 6–8 weeks, complete resolution of dyspnea and residual lung damage was observed in 33% and 25% of patients treated with oral steroids, respectively, while in the absence of oral steroids, resolution of dyspnea was observed in 40% and complete recovery of the lung parenchyma in 60% of patients. Thus, 80–90% of OP-like PCPS patients respond to steroid therapy, which is in good agreement with the initial response to oral corticosteroid therapy in other PIOPs [19,114,115,116].

A significant limitation of steroid therapy is the high risk of severe side effects. Oral steroid use increases the risk of sepsis (RR = 5.30; (3.80–7.41)), atrial fibrillation (RR = 3.75; (2.38–5.87]) and venous thromboembolism (3.33; [2.78–3.99]), as well as the incidence of bone fractures (1.87; (1.69–2.07)) [124,125]. In addition, all but occasional use of steroids is associated with exacerbations of pre-existing mental illness, mood disorders, steroid dementia, and akathisia [126]. The use of high doses of steroids for 1 or more months can lead to the development of acute psychosis and affective disorders [126]. One of the most severe complications is osteonecrosis, which occurs in 5–25% of patients receiving high doses of steroids for several months [127,128]. For example, the use of steroids to treat the consequences of SARS-CoV-1 infection resulted in osteonecrosis of the subchondral zone in 5% of patients [128].

The high relapse rate leads to the need to resume steroid therapy in about half of patients with OP [70]. Unfortunately, in the absence of clinical trial data, it is not possible to assess the likelihood of recurrence in OP-like PCPS. However, based on the published case studies, it can be concluded that the development of relapses is quite possible [129]. For example, a case of PCOP recurrence within 2 months after completion of a course of steroids was documented. Readministration of prednisolone at an initial dose of 20 mg/day with a gradual reduction over the following 6 months led to clinical and radiological improvement [129].

However, the development of resistance to steroid therapy appears to be the most serious issue. The emergence of resistance as a result of the PCOP relapse during the period of prednisolone dose reduction is described in [130]. A patient treated with steroids after discharge from the hospital experienced respiratory failure requiring rehospitalization, as well as the progression of OP [130]. Since the patient did not respond to steroid therapy, an attempt was made to use a combination of steroid and immunosuppressive therapy. Administration of mycophenolate mofetil (MMF) initially resulted in clinical improvement in the patient’s condition. Unfortunately, after 10 weeks of therapy, the patient was diagnosed with MMF-induced colitis and immunosuppressive therapy was discontinued [130].

This unprecedented “hidden” pandemic of OP-like PCPS clearly establishes the need for novel safe and effective therapies. It is therefore of paramount importance that full-fledged double-blind randomized trials be urgently conducted to evaluate the long-term safety and efficacy of steroid therapy for OP-like PCPS. Unfortunately, according to the ClinicalTrials.gov database, as of October 2022, there are only two active clinical trials dedicated to studying the use of steroids in OP-like PCPS (Table 2).

Oral corticosteroids constitute an effective therapy for OP-like PCPS with an expected response in 80–90% of patients. The high expected-relapse rate associated with the discontinuation of the therapy or a relapse of COVID-19, as well as the risk of the development of severe adverse reactions associated with long-term steroid use, imposes significant restrictions on the use of steroids for the treatment of OP-like PCPS. *We therefore propose that the use of corticosteroids should be limited to cases of disease progression and the presence of severe inflammation in the respiratory organs* [131]. (In the case of rapidly progressing pathology, the development of hypoxemia (SpO2 < 90%) can be used as a selection criterion for steroid therapy [130]. In milder cases, other criteria can be used, such as the deterioration of the gas-exchange function of the lungs (a decrease in DLCO by more than 15% from the baseline [131]) or an increase in the number and severity of fibrotic lesions on CT [129]).
pharmaceuticals-15-01574-t002_Table 2Table 2Clinical trials for the treatment of OP-like PCPS.
Treatment or InterventionStudy Design (Status)Enrollment and Key Inclusion Criteria(DLCO Impairment|CT Abnormalities|Inflammation|Respiratory Symptoms|Time since COVID-19)Objectives**Anti-inflammatory therapy**MethylprednisolonePO 0.5 mg/kg QD/1 mth (NCT04988282) [132]MC, RCT, Ph 4 (RECRUTING)642 patients in 2 groups(+ | + | − | + | ≥30 d)**Primary:** % of pts with mMRC = 0, FVC and CT imprv**Secondary:** % CT improvement, DLCO, FVC, SaO2, 6MWDPrednisone PO 20 mg QD/14 d(NCT04551781) [133]SC, SBRCT(COMPLETED)450 patients in 2 groups(− | + | − | −)**Primary:** improved resolution of CT infiltrates, <5%, 5–25%, and >25% infiltrates Prednisone PO QD/6 wk ^1^,PO 10 mg QD/6 wk (NCT04657484) [134]SC, RCT(COMPLETED)130 patients in 2 groups(+ | + | − | + | 3–8 wk)**Primary:** % of pts with ≥90% CT improvement**Secondary:** CT improvement (>50%, but <90%), FVC, SpO2, dyspnea score (mMRC), 1 m STS, 6MWD, KBILD, SF-36 Prednisone PO QD/24 wk ^2^,PO QD/12 wk ^3^,(NCT04534478) [135]SC, RCT, Ph 4(NOT YET RECRUITING)120 pts in 2 groups (ratio 1:1)(+ | + | − | +)**Primary:** change in DLCO**Secondary:** % of pts with DLCO <80%, 6MWD, CT, complications, mortalityMontelukastPO 10 mg QD/1 mth (NCT04695704) [136]MC, DBRCT, Ph 3 (RECRUITING)284 pts in 2 groups(− | − | CRP | mMRC | 1–12 mth)**Primary:** COPD Assessment Test Scale**Secondary:** 1 min sit-to-stand test; O2 desaturation; VAS; mortality; etc.TreamidPO 50 mg QD/1 mth (NCT04527354) [137]MC, DBRCT, Ph 2 (COMPLETED)60 pts in 2 groups (ratio 1:1)(DLCO < 80% | + | − | mMRC | 2–8 wk)**Primary:** % pts with FVC and/or DLCO improvement**Secondary:** change in 6MWD, mBDS, mMRC, FEV1, FVC, FEV1/FVC, DLCO, TLC, FRC, KBILD, rate of reduction in the lung damage (CT)TreamidPO 50 mg QD/1 mth (NCT05516550) [138]DBRCT, Ph 2/3 (NOT YET RECRUITING)412 pts in 4 groups (ratio 1:1:1:1)(DLCO < 80% | >10% | − | mMRC | 1–3 mth)**Primary:** % of pts with CT and 6MWT improvement**Secondary:** frequency of clinically significant change in DLCO, rate of clinically significant recovery of exercise tolerance (Borg, BDI/TDI, MFIS scores)ColchicinePO 0.5 mg BID/3 wk (NCT04818489) [139]SC, SBRCT, Ph 4 (COMPLETED)260 pts in 2 groups(− | + | − | − | −)**Primary:** % of participants with fibrosis**Secondary:** FVC and FEV1, C-reactive protein, ferritin, erythrocyte sedimentation rate, LDHBIO 300 (genistein)PO 1500 mg QD/3 mth (NCT04482595) [140]MC, DBRCT, Ph 2 (RECRUITING)66 pts in 2 groups (ratio 1:1)(− | + | + | + | <12)**Primary:** RAND 36 score**Secondary:** change in RHI, 6MWD, 30/60 sec chair stand**Antifibrotic therapy**NintedanibPO 150 mg BID/12 mth (NCT04541680) [141]SC, DBRCT, Ph 3 (RECRUITING) 250 pts in 2 groups(DLCO < 70% | ≥10% | − | − | −)**Primary:** change in FVC**Secondary:** DLCO, HRCT, Dyspnea, Biomarker assay (KL-6, NT-proBNP, CRP, D-dimers), etc.NintedanibPO 150 mg BID/6 mth (NCT04619680) [142]MC, DBRCT, Ph 4 (RECRUITING)170 pts in 2 groups (ratio 1:1)(DLCO < 80% | + | − | − |1 mth)**Primary:** change in FVC**Secondary:** HRCT, SGRQ, KBILD, LCQ, etc.NintedanibPO 150 mg BID/2 mth (NCT04338802) [143]RCT, Ph 2 (UNKNOWN STATUS)96 pts in 2 groups(− | + | − | − | −)**Primary:** change in FVC**Secondary:** DLCO, CT, 6MWTNintedanibPO 150 mg BID/6 mth (NCT04856111) [144]SC, SBRCT, Ph 4 (ACTIVE, NOT RECRUITING)48 pts in 2 groups(− | ≥10% | − | − | ≤4 mth)**Primary:** change in FVC**Secondary:** % of pts with composite response (mMRC score < 2, FVC ↑, improvement in SaO2 >92%); mMRC score, 6MWD, SaO2, HRCT scores, SF-36 score, KBILD score; FACIT-Dyspnea-10 scale; etc.Pirfenidone2400 mg QD/6 mth (NCT04856111) [144]LYT-100PO BID/3 mth (NCT04652518) [145]MC, DBRCT, Ph 2 (COMPLETED)185 pts in 2 groups(− | + | − | mMRC | −)**Primary:** change in 6MWD**Secondary:** change in Dyspnoea-12, SGRQ-I and mBDS scores, SF-36 score assessmentFuzheng HuayuPO 1600 mg TID/3 mth (NCT04279197) [146]SC, DBRCT, Ph 2 (COMPLETED)142 pts in 2 groups(− | + | + | − | ≥1 wk)**Primary:** % of pulmonary fibrosis and FVC impairment**Secondary:** 6MWD, % of pulmonary inflammation, clinical symptoms, QOL-BREF, PHQ-9, and GAD-7**Hypoxemia Management**S-1226inh 240 kPa for 90 min(NCT04842448) [147]DBRCT, Ph 2 (NOT YET RECRUITING)48 pts in 2 groups (ratio 1:1)(− | − | − | + | ≥ 1 mth)**Primary:** safety and tolerability**Secondary:** resp. symptoms (cough, breathlessness, 6MWD) and lung function (FEV1, SGRQ, FVC, DLCO, SpO2, mMRC, RPE, VAS)HBO2inh 3–4 min BID/1 wk (NCT04949386) [148]SC, DBRCT, Ph 2 (RECRUITING)80 pts in 2 groups (ratio 1:1)(− | − | − | + | ≥ 3 mth)**Primary:** RAND 36 score**Secondary:** change in RHI, 6MWD, 30/60 sec chair standOzone plasma (NCT05089305) [149]MC, Non-RT, Ph 2 (ENROLLING BY INVITATION)35 pts in 1 group(− | + | − | + | −)**Primary:** TLco, 6MWT and inflammation (CRP, TNFa, IL6)**Secondary:** EQ-5D and mMRC, CRP, D-dimer, etc.**Cell therapy**ExoFloIV at Day 0, 2, 4(NCT05116761) [150]DBRCT, Ph 1/2 (NOT YET RECRUITING)60 pts in 2 groups(− | − | − | + | 1–5 mth)**Primary:** increased 6MWD**Secondary:** EQ-5D and MRC scores, levels of C-reactive protein, D-dimer, atrial natriuretic peptidesCOVI-MSCIV at Day 0, 2, 4(NCT04992247) [151]DBRCT, Ph 2 (NOT YET RECRUITING)60 pts in 2 groups(− | − | − | + | ≥ 3 mth)**Primary:** change in 6MWD**Secondary:** change in 6MWD, pulmonary function, SpO2, etc.MON002Single infusion(NCT04805086) [152]SC, Non-RT, Ph 1/2 (RECRUITING)5 pts in 1 group(− | + | − | + | −)**Primary:** safety profile**Secondary:** change in FVC and TLCO; % of FVC ↓; time to ≥10% decrease in FVC; etc.^1^ PO 40 mg QD/1 wk, 30 mg QD/1 wk, 20 mg QD/2 wk, 10 mg QD/2 wk; ^2^ PO 0.75 mg/kg QD/4 wk, 0.50 mg/kg QD 4 wk, 20 mg QD/4 wk, 10 mg QD/6 wk, 5 mg QD/6 wk; ^3^ PO 0.5 mg/kg QD/3 wk, 20 mg QD/3 wk, 15 mg QD/2 wk, 10 mg QD/2 wk, 5 mg QD/2 wk. Clinical trials of Phases II, III, and IV were located on Clinicaltrials.gov using searching terms (“SARS-CoV” OR “COVID”) as a condition and (“organizing pneumonia” OR “post” OR “long” OR “sequelae” OR “persistent”) AND (“lung” OR “pulmonary” OR “respiratory” OR “interstitial”) as other terms. Only those considered PCPS were chosen. Clinical trials including using of dietary or food supplements, manipulating or behavioral therapy, investigation of devices, and aimed at the treatment of asthenia, pain, anxiety, or other cognitive and neurological pathology were not considered.

### 4.2. Anti-Inflammatory Therapy in Clinical Trials

It was previously demonstrated that key plasma inflammatory markers (IL-6 and CRP) were significantly elevated in individuals with ongoing pulmonary sequelae of COVID-19 developed after the resolution of the acute phase. This was associated with a higher proportion of SARS-CoV-2-specific T cells and correlated with reduced lung function and duration of dyspnea [153], thereby indicating that anti-inflammatory drugs should be tested as a potential therapy for PCPS.

**Colchicine**, an alkaloid extracted from autumn crocus, reduces NF-KB complex activity and decreases immune response and inflammatory genes, such as NALRP3, IL-1, Pro-IL-1β, IL-6, and TNF, and demonstrates high efficacy in numerous diseases characterized by systemic inflammation and uncontrolled activation of the innate immune response [154,155]. In a meta-analysis of five randomized clinical trials, colchicine was found to produce a statistically significant reduction of COVID-19 severity along with a decrease in CRP levels [156].

**Montelukast** is a cysteinyl leukotriene receptor antagonist. Leukotrienes and their receptors are mediators of tissue damage and concomitant inflammation and are well-known targets in many respiratory diseases. Montelukast was found to reduce levels of CRP, eosinophil cationic protein, IL-8, myeloperoxidase, and eosinophil counts, and increase levels of the anti-inflammatory cytokine IL-10 in adults with asthma [157,158]. Montelukast also showed anti-inflammatory activity in many animal models of various respiratory diseases. For example, montelukast was found to inhibit both acute lung injury induced by LPS in mice and LPS-induced human neutrophil activation [159]. In a retrospective study, it was shown that hospitalized COVID-19 patients treated with montelukast had fewer events of clinical deterioration [160]. A Phase III randomized controlled double-blind clinical trial testing montelukast in COVID-19 patients has been announced [161]. The trial’s endpoints are the severity of COVID-19 symptoms and the levels of inflammatory biomarkers. Montelukast has also partially resolved bronchoconstriction and had a considerable beneficial effect in patients with persistent post-COVID-19 cough [162].

**Treamid** is a bisamide derivative of a dicarboxylic acid. Treamid has exhibits anti-inflammatory and antifibrotic effects [113] and stimulates the regeneration of lung endothelium in animal models of pulmonary fibrosis. In a bleomycin-induced lung fibrosis model, administration of treamid resulted in the reduction of the amount of connective tissue, normalization of microvascular architecture, and restoration of parenchyma. Treamid was found to decrease in lung homogenates the number of lymphocytes and macrophages, the concentration of IL-13, as well as the levels of total collagen, collagen type 1, hydroxyproline, and fibronectin [163]. According to the results of a Phase II clinical trial in post-COVID-19 pneumonia patients, treatment with treamid resulted in clinically significant improvement, evidenced by an increase in forced vital capacity (FVC) and diffusing capacity for carbon monoxide (DLCO) as well as the reduction in dyspnea [138,164].

**BIO 300** is an isoflavone commonly found in soy and functions as a selective estrogen receptor-beta (ERb) agonist and was developed as an effective prophylactic for hematopoietic acute radiation syndrome [165]. In a guinea pig asthma model, treatment with BIO 300 resulted in a substantial reduction in pulmonary eosinophilia, ovalbumin-induced bronchoconstriction, and airway hyper-responsiveness. Based on the provided evidence of parallel etiologies between SARS-CoV-2 infection and radiation injury [166], BIO-300 may have potential in reducing COVID-19-related inflammation [167].

### 4.3. Antifibrotic Therapy in the Management of OP-like PCPS

Pulmonary fibrosis is considered to be one of the key concerns regarding COVID-19 pulmonary sequelae as it is associated with architectural distortion of the lung parenchyma and overall impairment of lung function resulting in decreased quality of life [168]. The antifibrotic agents pirfenidone and nintedanib are currently approved for idiopathic pulmonary fibrosis (IPF); however, treatment with these drugs is associated with frequent discontinuation because of adverse events during the treatment period and due to the progression of IPF or acute exacerbations [169]. This may also be a limitation for the prolonged treatment of post-COVID-19 sequalae.

**Pirfenidone** (5 methyl-1-phenyl-2-[1H] pyridone) inhibits the production of fibroblasts, suppresses the release of transforming growth factor beta (TGF-β1), inhibits the release and deposition of collagen [170], and reduces the production of proinflammatory cytokines such as TNF-α [171]. In the treatment of mild-to-moderate IPF with pirfenidone in adults, the dose is gradually increased from 800 mg/day up to 2400 mg/day over the course of 3 weeks. The meta-analysis of six randomized controlled trials (1073 subjects with pulmonary fibrosis) revealed that the pirfenidone group had a significantly higher rate of gastrointestinal (nausea, dyspepsia, diarrhea, and anorexia; RR = 2.11, 95% CI: 1.71–2.61, *p* < 0.001), neurological (dizziness and fatigue; RR = 1.68, 95% CI: 1.39–2.03, *p* < 0.001), and dermatological (photosensitivity and rash; RR = 2.88, 95% CI: 1.93–4.31, *p* < 0.001) adverse events compared to the placebo group [172]. The number of discontinuations in the pirfenidone group was also significantly higher compared to the placebo group. Given the presence of alveolar interstitial fibrosis in severe-COVID-19 patients, a Phase III clinical trial was conducted to evaluate the efficacy and safety of pirfenidone in 146 patients with severe COVID-19 [173,174]. No statistically significant difference between patients of the pirfenidone and placebo groups were found in the KBILD and CT images; however, some changes in CT scores (consolidation, GGO, reticulation) were observed. The levels of pulmonary inflammatory cytokines (IL-2R, TNF-a) were also significantly decreased [174], which confirmed the benefits of pirfenidone therapy as an anti-inflammatory treatment. The safety and side-effect profile were similar to that observed in patients with IPF. Another clinical trial tested pirfenidone in patients with COVID-19-induced severe acute respiratory distress syndrome (SARS) who also required mechanical ventilation [175]. In a randomized open-label trial comparing pirfenidone and corticosteroids in post-COVID-19 pulmonary fibrosis prevention in 60 patients with severe COVID-19 infection, it was shown that the final fibrosis score and proportion of patients who died at week 6 were statistically significantly lower in the pirfenidone group [176]. A case study of long-term pirfenidone therapy in a 66-year-old female patient with marked restrictive ventilatory dysfunction and impaired diffusing function showed diminishing fibrosis patterns on CT image and improvement in mMRC score, 6 min walking test distance, and pulmonary function (FVC, TLC, DLCO) over 96 weeks after treatment, indicating that pirfenidone may be an effective therapeutic strategy for post-COVID-19 pulmonary fibrosis [177]. A Phase II trial aimed to evaluate the efficacy of pirfenidone in patients with fibrotic lung sequelae after recovery from the acute phase of severe COVID-19 pneumonia [178].

**Nintedanib** is a multikinase inhibitor inhibiting the receptor tyrosine kinase of platelet-derived growth factor (PDGF), fibroblast growth factor (FGF), vascular endothelial growth factor (VEGF), and transforming growth factor-beta (TGFβ) [179], thus targeting several molecular pathways involved in the pathogenesis of IPF, such as fibroblast proliferation, migration and differentiation, and the secretion of ECM. Moreover, nintedanib exerts anti-inflammatory activity by inhibiting IL-1β [180]. The most frequent observed adverse effects of nintedanib in clinical trials, real-world clinical practice, and long-term extension studies were gastrointestinal (diarrhea) and cardiovascular [181]. In an interventional study of nintedanib treatment in patients with COVID-19 requiring mechanical ventilation, it was shown that the administration of nintedanib may offer potential benefits for minimizing lung injury in COVID-19, evidenced by the reduction of the mechanical ventilation duration and the percentages of high-attenuation areas on CT volumetry at the cessation of ventilation [182]. In a single-center study, the efficacy of nintedanib and pirfenidone in combination with steroids in the management of post-COVID-19 lung fibrosis was compared [183]. Significant improvement measured by the CT severity score at 12 weeks was found in the group receiving nintedanib. In a retrospective analysis, the continuity of pirfenidone and nintedanib treatments in IPF patients was compared [169]. It was found that within one year of the treatments (as well as throughout the observation period), the discontinuation rate due to adverse events was significantly higher in the nintedanib group than in the pirfenidone group.

**LYT-100** is a deuterated form of pirfenidone and an inhibitor of proinflammatory mediators, such as TNF-α, TGF-β, and IL-6, and is being studied for diseases that are associated with inflammatory and fibrotic pathologies, including lymphedema [184] and IPF [185].

**Fuzheng Huayu Tablet** [186] is a herbal product indicated for liver fibrosis. It was found to have a noticeable positive effect on pulmonary interstitial inflammation and fibrosis in bleomycin-induced rat models of fibrosis.

### 4.4. Hypoxemia and Pulmonary Hypertension as PCPS Targets

When injured host cells release damage-related inflammation mediators, immune cells are activated and endothelial cell damage ensues. This leads to aberrations in the function and structure of pulmonary vessels and, ultimately, hypoxia [187]. Hypoxia, in turn, activates endothelial, mesenchymal, and immune cells and promotes thrombotic fibrosis and epithelial–mesenchymal transition (EMT), which results in vascular remodeling. Similar processes are the hallmarks of pulmonary hypertension and chronic obstructive pulmonary disease (COPD). It was shown that post-COVID-19 patients may experience both restrictive pulmonary functional aberrations as well as bronchial obstruction [188]. Mitigation of hypoxia and bronchial obstruction, in turn, reduces endothelial cells injury, thereby preventing their transition to a defensive state. In another study, response to bronchodilation was evaluated in 105 consecutive patients soon after they were discharged after recovering from moderate-to-severe COVID-19 and less than 2 months after the onset of original symptoms. According to spirometric tests, a mixed obstructive–restrictive lung syndrome was present in the most patients (56%), indicating that treatment with bronchodilators may induce a functional improvement that, even if modest in magnitude, may noticeably facilitate breathing.

**S-1226** is a potent, safe, and well-tolerated gas mixture of 8% CO_2_ (natural bronchodilator) in perflubron (a synthetic surfactant). The inhaled S-1226 induces rapid bronchodilation in obstructed airways, which results in enhanced oxygenation of the blood, concomitant improved mucus clearance, restoration of alveolar surfactant function, and ultimately reduction of breathing effort. S-1226 was developed as a rescue therapy for mild atopic asthma [189] and was found to have potential beneficial effects in subjects with cystic fibrosis lung disease in a Phase IIA clinical trial [190].

**Hyperbaric oxygen treatment** [191] consists of exposure to 100% oxygen under higher-than-atmospheric pressure, which allows reaching tissues rapidly at elevated concentrations. Hyperbaric oxygen treatment reduces the expression of L-1β, IL-6, and TNF-α, was shown to be effective in the treatment of radiation injury [192] and wound-healing [193], and has been proposed as a treatment option to prevent the development of post-COVID-19 fibrosis [194].

**Ozone** is able to generate bioactive mediators acting on the complex cross-talk between oxidative stress, inflammation, and vascular function. Although the inhalation of ozone produces pulmonary toxicity, administration of ozone via appropriate routes and in small doses can lead to immunomodulation and anti-inflammatory response. Furthermore, it is inexpensive and easily available, and has therefore been suggested as adjuvant therapy in mild or moderate COVID-19 and a preventative against the progression to critical disease [195]. It was shown that oxygen–ozone immunoceutical therapy (3–5 cycles with 100–200 mL of 45 μg/mL O2-O3 mixture QD for 5 days) in fifty patients hospitalized with COVID-19 and acute respiratory disease syndrome (ARDS) significantly ameliorated major respiratory indexes, reduced levels of inflammatory markers, and ultimately reduced the number of patients needing intensive care unit (ICU) hospitalization (CRP, IL-6) [196]. Other clinical trials for the evaluation of the effect of ozone in patients with acute and postacute COVID-19 are currently ongoing [197]. In the initiated clinical trial [149], the ozone plasma (ppb < 0.07) will be administered for 5 min TID for 2 weeks.

### 4.5. Cell Therapy in the Management of OP-like PCPS

Mesenchymal stem cells (MSC) are multipotent stromal cells that have the ability to self-renew and exhibit multilineage differentiation. They are present in virtually all tissues and, depending on context, can differentiate into any cell type, such as, for example, lung epithelial cells [198]. Unfortunately, MSC treatments are in very early clinical development and no information regarding their clinical efficacy is available so far.

**ExoFlo**, bone-marrow-mesenchymal-stem-cell-derived extracellular vesicles contain growth factors and can downregulate inflammation and upregulate tissue repair [150].

**COVI-MSC** is allogeneic adipose-derived mesenchymal-stem-cell treatment. Adipose-derived mesenchymal stem cells secrete proangiogenic factors, inducing the proliferation of endothelial cells and promoting vascularization that might contribute to the improved outcomes observed during mesenchymal-stem-cell infusion in COVID-19 patients [199].

**MON002** is an autologous monocyte cell therapy (monocytes/macrophages passaged in vitro) with antifibrotic properties [152].

## 5. Conclusions

The rapid spread of COVID-19 led to a pandemic that poses a global threat not only to human health, but the very fabric of society as a whole. Patients who have recovered from COVID-19 may exhibit signs of fibrotic lung damage and the concomitant significant reduction in DLCO. These factors are frequently associated with severe PCPS. Persistent respiratory problems related to lung fibrosis may ultimately produce substantial population morbidity, long-term disability, and mortality due to the progression of OP. The incidence rate of OP-like PCPS can be estimated at 10–40% after severe and critical infection and 5–20% for moderate cases. Moreover, the incidence rate of persistent pathology can be estimated at 1–3% one year after hospitalization with acute COVID-19. Since the probability of getting severe COVID-19 in general population is about 5% over the period of 1 year, the prevalence of persistent OP-like PCPS is expected to be 6–20 cases per 10,000 people or 4–14 million people globally.

Despite substantial resources allocated worldwide to finding a solution to this problem, there are no approved treatments for PCOP or OP-like PCPS. Although the use of oral steroids provides a therapeutic response in most cases, the high toxicity and risk of relapse dramatically limit the applicability of this therapy. Based on the pathogenesis of PIOP and OP-like PCPS, we believe that treatment with anti-inflammatory agents in the first 6 months post-COVID-19 will minimize the residual lung inflammation and normalize the impaired lung-diffusion capacity. This should ameliorate respiratory symptoms, restore exercise tolerance, and boost lung tissue regeneration. We hope that the clinical trials discussed herein will result in the discovery of novel treatment options for the amelioration of OP-like PCPS and marked reduction of the risk or severity of pulmonary fibrosis.

## Figures and Tables

**Figure 1 pharmaceuticals-15-01574-f001:**
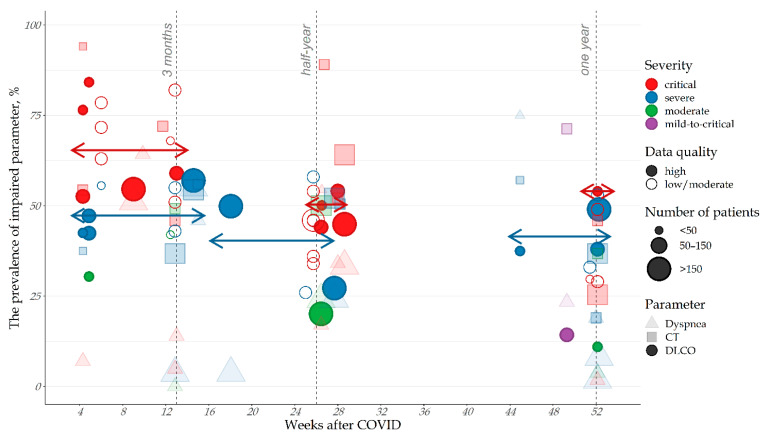
The prevalence of the pulmonary gas exchange impairment, depending on the severity of the disease and the time elapsed since the infection onset. (see Appendix A for more information).

**Figure 2 pharmaceuticals-15-01574-f002:**
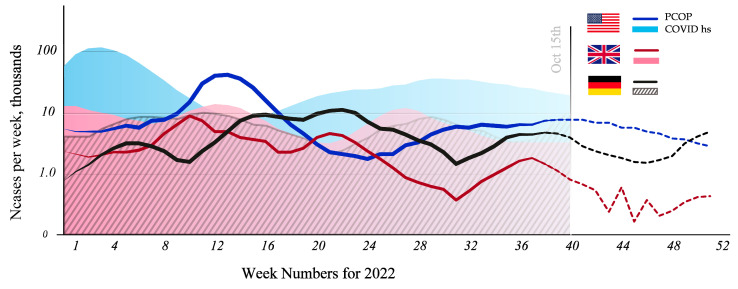
OP-like COVID-19 sequelae 2022 forecast in the USA, UK, and Germany.

**Figure 3 pharmaceuticals-15-01574-f003:**
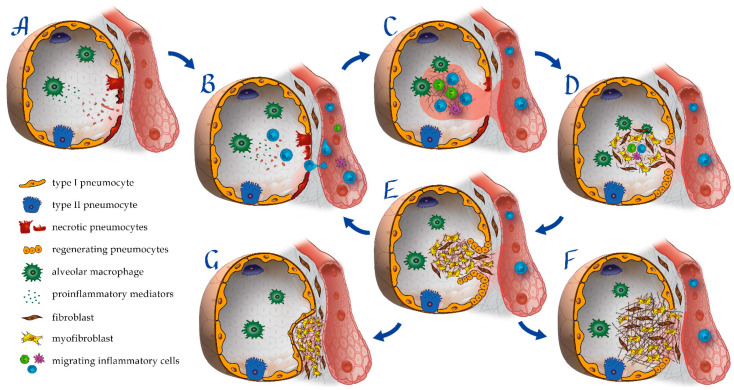
Pathogenesis of secondary PIOP and OP-like PCPS. (**A**) Virus-induced alveolar injury with alteration and necrosis of alveolar epithelial cells. (**B**) Aberrant inflammatory-induced chemotaxis of macrophages and neutrophils to the lung tissue. (**C**) Inflammatory-induced ROS production and epithelial and endothelial damage, vascular leakage with alveolar hemorrhage, and intra-alveolar edema. (**D**) Alveolar spaces colonization by pulmonary fibroblasts and myofibroblasts. Extracellular myxoid matrix deposition in the intra-alveolar space. (**E**) Progressive organization of intra-alveolar granulation tissue into mature fibrotic collagen-rich bundles. (**F**) Recurring cycles of alveolar epithelial cells damage and subsequent regeneration lead to gradually interlobular and intralobular septal thickening associated with bronchial dilations, suggesting the development of fibrosis. (**G**) The intra-alveolar buds are remodeled into the interstitium, and the collagen bundles (fibroblastic plugs) are covered with type I alveolar epithelial cells.

**Table 1 pharmaceuticals-15-01574-t001:** Etiology, pathogenesis, and clinical manifestation of OP-like PCPS, PIOP, and cryptogenic OP.

	OP-like PCPS	Secondary PIOP	Cryptogenic OP
Etiology	Virus-induced damage to epithelial and endothelial cells associated with an aberrant inflammatory response [43]		Unknown
Pathogenesis	Intrapulmonary-induced lung damage, vascular leakage, intra-alveolar edema, gas exchange impairment, hypoxemia, and subsequent pulmonary hypertension [49,52]. Tissue injury causes fibroblasts proliferation and transition into myofibroblasts, and the subsequent formation of extracellular matrix [53]. In turn, damaged lung tissue and the immune cells produce proinflammatory cytokines and chemoattractants, thereby maintaining the inflammatory process [54]. The formation of a stable pathological cycle, including the processes of tissue damage and regeneration over time, leads to the formation of stable fibrotic lesions.		
Clinical manifestation	Gas exchange impairment [37,55], dyspnea [7,56,57], dry cough [57,58], fatigue [7,57], fever [58]	Gas exchange impairment [13,59], dyspnea [13,60], dry cough [13,60], fatigue [59], *fever* [13,60]	
Inflammatory markers	Blood: Neutrophilia [61,62], lymphopenia [61,62,63], elevated CRP [45,46,47,61], TNF-a [45,46,47], ferritin [64,65], D-dimer [7] BAL: Neutrophilia [61,62], eosinophilia [62,66], hemorrhage [67,68,69]	Blood: Neutrophilia [13,60], lymphopenia [13,60], *eosinophilia* [13,60], elevated CRP [70], *TNF-a* [71], creatinine [13], D-dimer [72] BAL: Neutrophilia [13,60], eosinophilia [13,60], hemorrhage [73] elevated IL-12 and IL-18 levels [74]	
Radiological features	Peripheral bilateral predominant diffuse multifocal GGO and consolidation and reticular pattern, located primarily in the lower fields [22,23,75,76,77,78,79,80,81]	Subpleural and/or peribronchovascular bilateral multifocal consolidation located primarily in the middle and lower field [13,82,83,84,85,86]	
Fibrotic markers	Fibrinogen [7], LDH [61]	Fibrinogen [87], LDH [13]	
Histopathological findings	Intra-alveolar granulation tissue, interstitial lymphocyte [53,62] infiltration [62,88], and fibroblastic tissue proliferation [62,89]	Intra-alveolar granulation tissue with mononuclear cells, foamy macrophages infiltration [87,90], interstitial lymphocyte infiltration [90,91]	

Attributes highlighted in italics are present only in secondary PIOP.

## Data Availability

Not applicable.

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
