# Peer review of "The Hidden Pandemic of COVID-19-Induced Organizing Pneumonia"

_pharmaceuticals, 2022, doi:10.3390/ph15121574_

Round 1

Reviewer 1 Report

Dear authors,ùI have read the manuscript and I think that it is very well written. Data are of interest and the image as well as the tables are rilevant in the explanation of the manuscript.

I have only a minor question regarding the use of ozone in these patients. When and how it may be used ad at which dosage

Author Response

Dear Reviewer,

Thank you very much for evaluating the manuscript and for your comments. We have add information regarding the use of ozone therapy (see lines 510-524). The newly added fragments in the main text are colored in yellow. 

Thank you again.
Sincerely,
F. Novikov on behalf of all authors.

Reviewer 2 Report

The author is present a conventional review paper, rather than a systematic review, or organising pneumonia following coronavirus disease-19. In the context, this is an appropriate approach, and the outcome is an interesting and readable paper, despite (or perhaps because of) the fact that it is very long and comprehensive. The use of language is very good, and I can find extremely few errors of grammar.

 The Introduction does not have the usual description of virology of COVID-19, and I would suggest a short paragraph about coronaviruses in general, the SARS outbreak of 2002-4, and the mutant variants of SARS-CoV-2. This would help place later elements of the paper in context: omicron and delta appear without explanation in lines 85 and 86, and SARS-1 is briefly discussed in lines 201-5, and line 264.

I have a number of more specific constructive criticisms:

Line 97: "the" is redundant

Line 105: I suspect "number" is meant to be "proportion" or "percentage"

Table 1, histopathological findings: "granulation" should be changed to "granulation tissue"; I recognise that this is rather inconsistent, but the term "granulation" is often used (especially by surgeons) for the naked eye appearances of granulation tissue; the change would also avoid confusion with "granuloma", which I have also seen (and, at least once, leading to a major error in patient care)

Line 156: the term "collagen globules" has a particular meaning in this histopathology, different to that meant by the authors. Collagen globules may be seen in gastro-intestinal stromal tumours, and spherulosis of the breast, and are very different to the fibroblastic plugs seen in organising pneumonias. There is a fine description of the latter at the https://www.pathologyoutlines.com/topic/lungnontumorboop.html

Line 189: Reference 63 is a case report, not showing particularly good evidence of chronic interstitial lymphocytic infiltration; more importantly, lymphocytic infiltration is a relatively minor component of septal thickening, when compared to oedema and fibrosis

Line 296: The proposal about the limitation of use of steroids for severe respiratory inflammation is entirely realistic and justifiable, but some mention of the criteria to be used to establish severe inflammation would be useful

Lines 427-9: Chronic obstructive pulmonary disease is very different to this post-COVID organising pneumonia, I do not think it is realistic to generalise improvement in the former to justify using the same medication for the latter

Lines 472-84: The authors describe three forms of cell therapy, but provided nothing about the outcomes of the studies they refer to. I appreciate that this is because these are trials registered at a clinical trials database, for which there no current published outcome, but some reference might be made to this in the text so that the reader does not need to make this inference

Line 505: As a pathologist, I am pessimistic about the possibility of preventing pulmonary fibrosis altogether, but optimistic about reducing its severity; perhaps a change to “…reduce the risk or severity of…” should be considered.

References: There are many deviations from the recommendations at https://mdpi-res.com/data/mdpi_references_guide_v5.pdf; in particular, many of the citations are limited to the name of the first author, followed by “et al”. There are a lot of other differences from the recommendations in the style guide, too. The list of references requires comprehensive revision by the authors.

References, clinicaltrials.gov: I have looked at several of these, and the titles quoted by the authors are often different from the titles in the originals; these should be reviewed by the authors.

Reference 26: This is incomplete. I suspect it is meant to include https://institute.global/policy/hidden-pandemic-long-covid

Reference 81: This appears to include part of an abstract, and lacks the name or names of authors.

Author Response

Dear Reviewer,

Thank you very much for evaluating the manuscript and for your comments. We agree with this remark corresponding changes have been made in the manuscript (Please see the detailed description in attachment). The newly added or edited fragments in the main text are colored in yellow.

Thank you again.
Sincerely,
F. Novikov on behalf of all authors.

Round 2

Reviewer 2 Report

No comment